# Progress of Antimicrobial Mechanisms of Stilbenoids

**DOI:** 10.3390/pharmaceutics16050663

**Published:** 2024-05-15

**Authors:** Xiancai Li, Yongqing Li, Binghong Xiong, Shengxiang Qiu

**Affiliations:** 1Key Laboratory of Plant Resources Conservation and Sustainable Utilization, South China Botanical Garden, Chinese Academy of Sciences, Guangzhou 510650, China; xiongbh@scbg.ac.cn; 2Key Laboratory of South China Agricultural Plant Molecular Analysis and Genetic Improvement, South China Botanical Garden, Chinese Academy of Sciences, Guangzhou 510650, China; liyongqing@scbg.ac.cn

**Keywords:** stilbenoids, antimicrobial mechanisms, conventional targets, antibiofilm, antivirulence, reversing drug resistance

## Abstract

Antimicrobial drugs have made outstanding contributions to the treatment of pathogenic infections. However, the emergence of drug resistance continues to be a major threat to human health in recent years, and therefore, the search for novel antimicrobial drugs is particularly urgent. With a deeper understanding of microbial habits and drug resistance mechanisms, various creative strategies for the development of novel antibiotics have been proposed. Stilbenoids, characterized by a C_6_–C_2_–C_6_ carbon skeleton, have recently been widely recognized for their flexible antimicrobial roles. Here, we comprehensively summarize the mode of action of stilbenoids from the viewpoint of their direct antimicrobial properties, antibiofilm and antivirulence activities and their role in reversing drug resistance. This review will provide an important reference for the future development and research into the mechanisms of stilbenoids as antimicrobial agents.

## 1. Introduction

Microorganisms are life forms with the widest distribution, the most abundant species and the largest biomass on earth. Some microorganisms, such as intestinal beneficial microorganisms, industrial microbial strains and agricultural microbial agents, are widely used in the human health industry and agricultural production and have greatly promoted the development of the economy and society. Others, however, cause pathogenic diseases. Bacterial infections were fatal until the 1940s, when penicillin was discovered. In the years that followed, the discovery of antibiotics released people from the hegemony of infection and improved our quality of life and life expectancy. However, with the abuse of antimicrobial drugs, many new challenges in the treatment of pathogenic infections have emerged recently. First of all, new pathogens continue to emerge, such as SARS-CoV and SARS-CoV-2, and there is still a lack of effective drugs against these pathogens. Second, the development of drug resistance makes it difficult to treat previously curable infections. According to statistics, an estimated 4.95 million people died from bacterial AMR in 2019, of which 1.27 million died directly [1]. Third, many existing antimicrobials have defects. For example, the antifungal drugs azole, polyene and echinomycin can cause a variety of side effects in the human body. Therefore, it is urgent to find more efficient and safer antimicrobials.

Stilbenoid, characterized by a C_6_–C_2_–C_6_ carbon skeleton, possess a wide range of pharmacological activities. For example, resveratrol has demonstrated anti-inflammatory, antioxidant, hypolipidemic, hypoglycemic and anticancer activities [2,3,4]. It can also be used as a supplement to prevent cardiovascular system diseases, nervous system diseases and cancer [5]. In addition, the antimicrobial activity of stilbenoids is also of great concern to phytochemists and pharmacologists. Plant-derived stilbenoids are a class of important phytoalexins produced by plants to protect against pathogen infections and toxins. Mattio et al. summarized the sources, chemical structures and the antibacterial mechanism of natural stilbenoids [6]. The role of stilbenoid-abundant extracts in the mitigation of mycotoxins in food and feedstuffs has also attracted great interest [7]. However, there is no review providing a comprehensive overview of the antimicrobial (antibacterial and antifungal) mechanisms of stilbenoids from the perspective of their direct microbicidal activity, antibiofilm activity and/or regulation of the pathogenicity of microbes. Here, this review will emphasize recent achievements in these aspects of stilbenoids. 

## 2. The Family of Stilbenoids

Resveratrol was the first stilbenoid to be discovered. It was originally isolated from the root of *Veratrum grandiflorum* in 1939 by Takaoka [8]. Subsequently, with the introduction of more accurate metabolite identification/isolation techniques and chemical synthesis, more and more complex stilbenoids were identified. These stilbenoids can be divided into three categories according to their chemical structure, including monomers and oligomers (Figure 1). 

Monomeric stilbenes, such as resveratrol, piceatannol, pterostilbene and pinosylvin, are characterized by two phenyl rings, joined by an ethylene bridge. Oligomeric stilbenes are a group of compounds polymerized from monomeric stilbene. Resveratrol oligomers, which consist of two to eight or more resveratrol subunits, are the largest group of the oligomeric stilbenes [9]. Hopeaphenol, a tetramer of resveratrol isolated from *Hopea odorata* and *Balanocarpus heimti*, was the first oligomeric stilbenoid to be discovered in 1951 [10]. Subsequently, more and more oligomeric stilbenes from plants were identified. Oligomers can be polymerized from either homogenous or heterogeneous monomeric stilbenes. For example, hopeaphenol and ε-viniferin are polymerized from resveratrol units, while Gnetuhianin Q is a mixed dimer that is polymerized from both isorhapotigenin and resveratrol. Gnetuhainin K is the mixed dimer of gnetol and isorhapotigenin [11]. Those oligomers can be divided into benzofuran oligomers and benzocyclopentane oligomers. 

Bibenzyls (e.g., amorfrutin B and erianin, etc.) are comprised of two aromatic rings linked by an ethyl bridge. Monomeric bibenzyls can be polymerized into bisbibenzyls. However, no oligomers polymerized by more than three bibenzyl units have been found. 

Phenanthrenoids include the phenanthrenes, 9,10-dihydrophenanthrenes and their oligomers. More than one hundred dimers have been identified [12]. However, only three triphenanthrenes have been reported [13,14,15].

## 3. Structural Diversity and Antimicrobial Activity of Stilbenoids

Simple monomeric stilbenes (including resveratrol, piceatannol and pinosylvin) or bibenzyls (including 5-hydroxy-lunularic acid and 3,3′,5-trihydroxybibenzyl), synthesized through the phenylpropanoid pathway, are the precursor of many other stilbenoids in plants. It is generally believed that phenanthrenes and 9,10-dihydrophenanthrenes are formed by the oxidative coupling of the aromatic rings of stilbene and bibenzyl precursors, respectively [16,17]. However, other views on the biosynthesis of phenanthrenoids have been discussed in another review [16]. Subsequently, the diversity of stilbenoids is greatly increased via enzymatic modifications of side groups on the precursors, such as methylation, prenylation, oligomerization, hydroxylation, glycosylation and isomerization. The chemical diversity of stilbenoids provides them with a wide range of antimicrobial activities (Table 1), especially drug-resistant microbes.

### 3.1. Methylation

Methylation is the most common modification in plants. Methylation alters the chemical properties of hydroxyl stilbenes and therefore affects their antimicrobial activity. For example, rapid absorption and metabolism result in the limited oral bioavailability of resveratrol, which restricts the application of this potentially valuable compound in clinical trials [18,19]. However, pterostilbene, a dimethoxy derivative of resveratrol, shows better bioavailability and stronger antifungal activity [20]. In addition, stilbenes such as pinosylvin, piceatannol and pinostilbene, which contain a varied number of methylated groups, also exhibit higher antimicrobial activity compared to resveratrol [21,22,23,24]. Li et al. shows that the chemical affinity balance of substituent groups of stilbenes is key for their antifungal activity [25].

### 3.2. Isomerization

Structurally, stilbenes can exist as *cis* (*Z*) and *trans* (*E*) isomers, based on the configuration of the ethylene bridge, such as *trans*-resveratrol and *cis*-resveratrol. Generally, the two isomers exhibit different chemical characteristics and thereby different biological activities. The *E*-type configuration is more common in nature. This configuration is not sterically hindered, being therefore more stable. For example, research showed that *trans*-resveratrol exhibited stronger antibacterial effects compared to *cis*-resveratrol [26].

### 3.3. Prenylation

Prenyl groups can be attached to the skeleton at different positions and in different configurations (e. g., chain or ring-closed prenylation). One isoprenyl (3,3-dimethylallyl) moiety bound to the aromatic ring of stilbenes is the most common form in nature. In addition, prenyl patterns such as 3-methyl-but-1-enyl and geranyl are also often introduced to various positions of the stilbene backbone, along with oxidation, methylation and cyclization moieties, increasing the structural diversity of stilbenoids. Prenylation of compounds improves the binding affinity to receptors and is beneficial to membrane partitioning, consequently enhancing bioactivity [27]. For example, prenylated stilbenoids generally exhibit stronger antibacterial activity with MICs (minimal inhibitory concentration) in the µg/mL range. Improved properties, such as hydrophobicity, charge or molecular geometry, may contribute to their enhanced activity [28]. However, the rule is not absolute. For example, isopentenylated longistylin C shows lower antifungal activity than pinosylvin monomethyl ether (PME) [25]. This is closely related to their mechanism of action.

### 3.4. Oligomerization

Oligomeric stilbenoids are formed by oxidative coupling between homogenous or heterogeneous monomeric stilbenoids. A series of oligomeric stilbenoids has been identified in recent years, of which some are superior in bioactivity, stability and selectivity compared to the parental monomers [29,30]. For example, tapinarof is a stilbene drug isolated from the gammaproteobacterial *Photorhabdus* genus [31]. Duotap-520, isolated from *Photorhabdus* gammaproteobacteria, is the dimer of tapinarof and exhibits enhanced activity against Gram-positive bacteria (methicillin-resistant *Staphylococcus aureus* and vancomycin-resistant *Enterococcus faecalis*), with no resistance development even after daily non-lethal exposure for a duration of three months [31]. Interestingly, this metabolic dimer of tapinarof has gained antibacterial activity but has lost its anti-inflammatory activity [31]. Another dimer is dehydro-δ-viniferin, which shows enhanced antibacterial activity compared to monomer resveratrol [32].

**Table 1 pharmaceutics-16-00663-t001:** Antimicrobial activities and mechanisms of stilbenoids.

Targets	Compounds	Details	Determination Methods	Refs
Cell membrane	Longistylin A	MRSA, MIC = 1.56 µg/mL; disturbing membrane potential and increasing permeability.	Micro-well dilution.	[33]
	Toremifene	*P. gingivalis* and *S. mutans*, MICs = 12.5–25 µM; disrupting the cell membrane.	Micro-well dilution.	[34]
	Dehydro-δ-viniferin	*L. monocytogenes*, MIC = 2 μg/mL.	Micro-well dilution.	[35]
	Pterostilbene	*F. nucleatum*, MIC = 20 µg/mL utilizing 2-hydroxypropyl-β-cyclodextrin as a solubilizer.	Micro-well dilution.	[36]
	Resveratrol	*S. aureus* ATCC 25923, MIC = 512 μg/mL; *P. aeruginosa* ATCC 27853, MIC > 512 μg/mL.	Micro-well dilution.	[32]
	Cajaninstilbene acid derivative 5b	*S. aureus* ATCC25923, MIC = 4 μg/mL; *S. epidermidis* ATCC12228, MIC = 1 μg/mL; *B. subtilis* ATCC6633, MIC = 0.5 μg/mL; interferring in PG synthesis pathway by targeting PgsA.	Micro-well dilution.	[37,38]
	PME	*A*. *flavus*, IC50 = 260 μg/mL; binding the phospholipids of cell membrane.	Agar drug plate growth assay.	[25]
Cell wall	Duotap-520	MRSAs, MICs = 4 μM; VRE, MIC = 6 μM; binding to lipid II.	Micro-well dilution.	[39]
	GW458344X	Inhibiting MurC activity, IC_50_ = 368 μM; MurD, IC50 = 104 μM; MurE, IC_50_ = 49 μM; MurF, IC_50_ = 59 μM.	Enzyme activity test.	[40]
	135C	*S. aureus*, MICs = 0.12–0.5 μg/mL; targeting cell wall teichoic acids.	Micro-well dilution.	[41]
	Plagiochin E	Inhibiting the activity of chitin synthases.	Enzyme activity test.	[42]
	Tamoxifen	*S. pombe*, MIC = 32 μg/mL; inhibiting Ccr1 NADPH-cytochrome P450 reductase activitie.	Micro-well dilution.	[43]
DNA	Resveratrol-trans-dihydrodimer	*B. cereus*, MIC = 15.0 μM; *L. monocytogenes*, MIC = 125 μM; *S. aureus*, MIC = 62.0 μM; *E. coli*, MIC = 123 μM, upon addition of the efflux pump inhibitor; inhibiting DNA gyrase.	Micro-well dilution.	[44]
	Triazolyl-pterostilbene derivative 4d	MRSAs, MICs = 1.2–2.4 μg/mL, MBCs = 19.5–39 μg/mL; inhibiting the activity of DNA polymerase.	Micro-well dilution.	[45]
	Oxyresveratrol	*C. albicans* ATCC90028, MIC = 5.0 μg/mL; *C. parapsilosis* ATCC22019, MIC = 5 μg/mL; inflicting cleavage on DNA.	Micro-well dilution.	[46]
	Resveratrol	*S. typhimurium*, MIC = 5 μg/mL; inducing DNA disruption.	Micro-well dilution.	[47]
Mitochondria	Resveratrol	*C. albicans* ATCC 90028, MIC = 20 µM; inducing mitochondria-dependent apoptosis.	Micro-well dilution.	[48]
	Plagiochin E	*C. albicans* CA2, MIC = 16 μg/mL; inducing mitochondria-dependent apoptosis.	-	[49,50]
ATPase	Resveratrol and piceatannol	Inhibiting ATPase activity; piceatannol, IC_50_ = 14 μM; resveratrol, IC_50_ = 94 μM.	Enzyme activity test.	[51]
Cell-division	Resveratrol	*E. coli*, MIC = 456 μg/mL; preventing Z-ring formation.	-	[52]
PTS system	Cajaninstilbene acid	Sensitive *Enterococcus* strains and VRE strains, MICs = 0.5–2 μg/mL; inhibiting carbohydrate specific type II transporters of PTS system.	Micro-well dilution; proteomics; q-PCR.	[53]
Calmodulin–calcineurin pathway	Tamoxifen	*S. pombe*, MIC = 35 μg/mL; *Candida* spp. and *C. neoformans*, MICs = 8–64 μg/mL; directly binding to calmodulin.	Agar drug plate growth assay; micro-well dilution	[54,55,56]
Virulence factors	Erianin	*S. aureus* ATCC25904, MIC = 512 μg/mL; inhibiting the activity of SrtA with a IC_50_ = 20.91 ± 2.31 μg/mL.	Micro-well dilution.	[57]
	Resveratrol	Reducing the secretion of α-hemolysin by downregulating saeRS.	Red blood hemolysis assay.	[58,59]
	DIDS	Inhibiting *V. vulnificus* toxicity to HeLa cells at 10–300 μM with no influence on host cell viability and bacterial growth; reducing the expression of *TolCV1*.	-	[60]
	Raloxifene	Enhancing the survival percentage of *C. elegans* infected with *P. aeruginosa* PA14 at 12.5–100 μg/mL; inhibiting pyocyanin production by binding and inhibiting PhzB2.	-	[61]
	(-)-Hopeaphenol	Reducing cell entry and subsequent intracellular growth of bacteria; inhibiting the expression of *YopE* with a IC_50_ = 8.8 μM; inhibiting the activity of YopH with a IC_50_ = 2.9 μM.	A luminescent reporter-gene assay (*YopE*); an enzyme-based YopH assay.	[62,63]
	Hopeaphenol, isohopeaphenol, kobophenol A and ampelopsin A	Reducing the pathogenicity of *P. syringae* pv. tomato DC3000 on leaves; reducing the expression of *hrpA*, *hrpL* and *hopP1* genes without influence on bacterial growth.		[64,65]
Biofilm	Resveratrol	Inhibiting swarming of *P. mirabilis* at 15 μg/mL, and completely inhibited swarming at 60 μg/mL.	-	[66]
		*S. typhimurium* SL1344, MIC > 512 μg/mL; inhibiting adhesion of *S. typhimurium* to HeLa cells; downregulating the expression of flagella genes.	-	[67]
		*P. gingivalis*, MICs= 78.12–156.25 μg/mL; reducing the expression fimbriae genes.	-	[68]
		*V. cholerae*, MIC = 60 μg/mL; antibiofilm at 10–30 μg/mL; inhibiting the activity of AphB.	Crystal violet assay; confocal laser scanning microscopy.	[69]
		*S. mutans*, MIC = 800 μg/mL; inhibiting biofilm formation at 50–400 μg/mL; reducing the biosynthesis of polysaccharide.	Growth curve assay; crystal violet assay; confocal laser scanning microscopy.	[70,71]
		Inhibiting pyocyanin production of *P. aeruginosa* PAO1 by directly binding LasR.	-	[72]
		Reducing MN production.	-	[73]
	Resveratrol and oxyresveratrol	Antibiofilm at 10 μg/mL and 100 μg/mL; reducing fimbriae production and the swarming motility of UPEC.	Crystal violet assay; confocal laser scanning microscopy.	[74]
	Trans-stilbene	Reducing biofilm of *S. aureus* ATCC6538 at 50–200 μg/mL; decreasing the expression of intercellular adhesion locus.	Crystal violet assay; confocal laser scanning microscopy.	[58]
	Piceatannol	*S. mutans*, MIC_50_ = 564 ± 38 μM; antibiofilm, IC_50_ = 52 ± 6 μM; inhibiting the activity of GtfB and GtfC.	Crystal violet assay; confocal laser scanning microscopy.	[75]
	Oxyresveratrol	*S. mutans*, MIC = 500 μg/mL; reducing biofilm formation at 62.5–250 μg/mL; suppressing the expression of *gtfB* and *gtfC*.	Micro-well dilution; confocal laser scanning microscopy.	[76]
	Amorfrutin B	Antibiofilm activity against *P. aeruginosa* PAO1 with a biofilm inhibition ratio of 50.3 ± 2.7 at 50 μM; binding the receptors of signal molecule.	Crystal violet assay.	[77]
	Cajaninstilbene acid analogue 3o	Inhibiting biofilm formation of *P. aeruginosa* with inhibition ratio of 49.50 ± 1.35% at 50 μM; suppressing the expression of *lasB* and *pqsA*.	Crystal violet assay.	[78]
	Riccardin D	Inhibiting biofilm formation at 16 µg/mL and 64 µg/mL using central venous catheter (CVC)-associated *C. albicans* biofilms in an infectious rabbit model; reducing the expression of hypha-specific genes.	XTT reduction assay; scanning electron microscopy; confocal laser scanning microscopy.	[79,80]
	Pterostilbene	Inhibiting the formation of *C. albicans* biofilms in vitro at 1–32 μg/mL; downregulating the expression of filamentation-related genes.	XTT reduction assays; confocal laser scanning microscopy; scanning electron microscopy.	[81]
Reversing antibiotic resistance	Resveratrol	Enhancing the efficacy of aminoglycosides against Gram-positive pathogens; inhibiting the activity of ATP synthase.	-	[82]
		Increasing susceptibility of *P. aeruginosa* PAO1 biofilm to aminoglycosides; inhibiting the expression of signaling molecule synthase genes *lasI* and *rhlI.*	-	[83,84]
	Pterostilbene	Restoring the effectiveness of meropenem against NDM-expressing strains; inhibiting NDM-1 hydrolysis activity at 4–32 μg/mL.	-	[85]
	Cajaninstilbene acid	Restoring the susceptibility of polymyxin B to *mcr-1* positive Gram-negative bacteria; inhibiting the enzymatic activity of MCR-1.	-	[86]
	Resveratrol, pterostilbene, and pinosylvin	Increasing sensitivity of *A. butzleri* strains to chloramphenicol, erythromycin and ciprofloxacin by acting as EPIs.	-	[87,88,89]
	Piceatannol	Increasing sensitivity of *S. aureus* to ciprofloxacin by decreasing PMF.	-	[90]

## 4. Antimicrobial Mechanisms of Stilbenoids

Stilbenoids exert antimicrobial activities via various modes (Figure 2). The antimicrobial mechanisms mainly include (I) acting on conventional targets, (II) acting on nonconventional targets and (III) reversing antibiotic resistance. Firstly, stilbenoids can inhibit conventional targets, such as the cell membrane, cell wall, DNA, the calmodulin–calcineurin pathway, mitochondria, cell division and the phosphoenolpyruvate (PEP)-dependent phosphotransferase system. Secondly, stilbenoids can function as antibiofilm or antivirulence agents. Stilbenoids inhibit biofilm formation and assist in the eradication of preformed biofilms. Moreover, stilbenoids can reverse drug resistance by inhibiting alternative targets, target-modifying enzymes or antibiotic-modifying enzymes.

### 4.1. Direct Antimicrobial Mechanisms of Stilbenoids

#### 4.1.1. Targeting the Cell Membrane

The antimicrobial mechanism elucidates the membrane-disruption effects of stilbenoids, resulting in the leakage of cell contents and subsequent cell death. For example, the anti-MRSA (methicillin-resistant *Staphylococcus aureus*) activity of longistylin A, a stilbene isolated from the leaves of *Cajanus cajan* (L.) Millsp, is mediated by disturbing the bacterial membrane potential and increasing permeability [33]. Li et al. showed that PME exerted anti-*Aspergillus flavus* activity by binding the phospholipids, leading to decreased fluidity and increased permeability of the cell membrane [25]. In addition, toremifene (an FDA-approved anticancer agent synthesized in 1981), pterostilbene, dehydro-δ-viniferin (a stilbene dimer chemically semi-synthesized from resveratrol), resveratrol and so on, are all shown to exert antimicrobial activity by disrupting the cell membrane [32,34,36].

Membrane-associated protein PgsA catalyzes glycerolphosphate to replace cytidine monophosphate to produce phosphatidylglycerol phosphate (PG-P). Subsequently, PG-P is dephosphorylated by PgpP to yield phosphatidylglycerol (PG), a major component of cell membrane [91]. Research has shown that a chemically synthesized cajaninstilbene acid-derivative, 5b, interferes in the PG synthesis pathway by targeting PgsA [37]. 

#### 4.1.2. Targeting the Cell Wall

The bacterial cell wall is a peptidoglycan polymer network that is composed of N-acetylglucosamine (GlcNAc) and N-acetylmuramic acid (MurNAc), with a pentapeptide attached. Firstly, the UDP-MurNAc-pentapeptide is coupled with bactoprenyl-phosphate (lipid I) on the cytosolic side of the cell membrane, followed by the coupling of the GlcNAc sugar by the enzyme MurG to produce lipid II. Next, lipid II, containing the complete peptidoglycan subunit, is translocated to the outside of the cell membrane. The subunits are inserted into the cell wall by penicillin-binding proteins (PBPs) to make the cell wall grow further. Finally, the lipid anchor is returned to the cell for the next round of synthesis [92]. Duotap-520, a natural stilbene dimer isolated from *Photorhabdus*, can bind to lipid II, preventing the role of lipid II [39]. Bacterial peptidoglycan biosynthesis is catalyzed by a series of Mur ligases in the intracellular steps. Research has shown that chemically synthesized aza-stilbene GW458344X is a competitive inhibitor of MurD, an enzyme catalyzing the reaction from UDP-MurNAc-Ala to UDP-MurNAc-dipeptide [40]. Wall teichoic acids (WTAs) are anionic glycopolymers anchored in the cell walls of Gram-positive bacteria and have critical functions in bacterial physiology. The first-committed step of WAT biosynthesis is catalyzed by TagA, a membrane-associated glycosyltransferase [93]. The two-component transporter TagGH is responsible for the exportation of WATs [94]. A mutation in genes (*tagH*, *tagA* and *tagG*) has conferred drug resistance to *S. aureus* against the chemically synthesized tris-stilbene 135C. Therefore, 135C showed promising activity against Gram-positive bacteria, probably via targeting cell wall teichoic acids [41].

The fungal cell wall, a complex matrix with no mammalian counterpart, plays a role in maintaining cell shape and integrity, and therefore presents an ideal drug target. The cell wall consists of complex components, mainly polysaccharides (glucans, mannans and chitins), proteins and lipids. Studies showed that the macrocyclic bis(bibenzyl) plagiochin E caused damage in cell wall of *C. albicans* by inhibiting chitin synthetase activities [42]. The NADPH-cytochrome P450 reductase Ccr1 plays a role in the cell wall assembly of budding yeast [95]. The deletion of Ccr1 could cause defects in yeast cell wall integrity. Tamoxifen, an estrogen receptor antagonist used for treating breast cancer, has been found to inhibit Ccr1 NADPH-cytochrome P450 reductase activities of fission yeast and *C. albicans* in a dose-dependent manner, thereby causing defects in cell wall integrity [43]. 

#### 4.1.3. Targeting the DNA

Stilbenoids can inhibit the synthesis of DNA. For example, resveratrol-trans-dihydrodimer, obtained by the oligomerization of resveratrol catalyzed by soybean peroxidase in vitro, has been shown to inhibit DNA synthesis by reducing DNA gyrase activity via blocking the ATP-binding site of the enzyme [44]. Another study demonstrated that the chemically synthesized triazolyl-pterostilbene derivatives (**4d**, **7d** and **7e**) exhibited potent anti-MRSA activity by inhibiting the activity of DNA polymerase [45]. Stilbenoids can also damage DNA. For example, oxyresveratrol inflicted cleavage on DNA by directly binding to the DNA, which in turn led to mitochondria-mediated apoptosis in *C. albicans* [46]. Resveratrol induces DNA disruption via pro-oxidant activity (including increasing ROS and malondialdehyde accumulation and depleting glutathione) against *Salmonella typhimurium* [47]. 

#### 4.1.4. Targeting the Mitochondria

Mitochondria not only power life via their varied metabolic functions, but also play a central role in apoptotic cell death. Mitochondrial membrane permeabilization upon stimulation usually causes cytochrome c release from the mitochondria and subsequent metacaspase activation, and eventually commits a cell to die [96]. As such, targeting mitochondrial membrane to manipulate cell death holds tremendous antimicrobial potential. Studies have demonstrated that resveratrol caused the loss of mitochondrial membrane potential (ΔΨm), leading to metacaspase activation, cytochrome c release and eventually apoptosis of *C. albicans* [48]. Another study showed that plagiochin E could induce mitochondria-dependent apoptosis in yeast, mainly including the activation of F(0)F(1)-ATPase, inhibition of dehydrogenase and release of cytochrome c, leading to metacaspase activation [49,50]. 

#### 4.1.5. Actions on other Conventional Targets

The cell membrane and wall, the DNA and mitochondria mentioned above are all conventional antimicrobial targets. In addition, stilbenoids can also act on other conventional targets. For example, resveratrol and piceatannol inhibited both the ATPase activity and ATP synthesis of *Escherichia coli* reversibly [51]. Resveratrol can also exert antibacterial activity by downregulating *FtsZ* expression and preventing Z-ring formation. FtsZ can polymerize to form a dynamic ring (Z-ring) to promote cell division in prokaryotes. The Z-ring plays a role in recruiting division-related proteins and directing septal peptidoglycan synthesis to initiate division [52,97].

The bacterial phosphoenolpyruvate (PEP)-dependent phosphotransferase system (PTS) consists of the coupled carbohydrate-specific transporters (called “EIIs”) and two general components, EI and HPr, encoded by *ptsI* and *ptsH*, respectively [98]. The PTS catalyzes the coupled phosphorylation of carbohydrates and their transport into cells. Firstly, the phosphoryl groups derived from PEP are transferred from the EI to an EII with the mediation of HPr. Finally, the phosphoryl groups are transferred to a carbohydrate substrate by its cognate EII [53]. Given that this system plays an important role in catalytic transport and has extensive regulatory functions, it presents an important drug target for the development of antibacterial agents [99]. Cajaninstilbene acid has been shown to inhibit vancomycin-resistant *Enterococcus* by inhibiting the carbohydrate-specific type II transporters of the PTS system, as confirmed by proteomics and fluorescence quantitative PCR [53]. 

Tamoxifen is a known inhibitor of mammalian calmodulin. Recent studies have found that tamoxifen could exert antifungal activity through targeting the calmodulin–calcineurin pathway. Evidence suggests that tamoxifen could induce a consistent phenotype in *C. albicans*, resembling yeasts that have lost the calmodulin function (e. g., disrupting cell integrity, blocking new bud emergence, interfering with the polarization of the actin cytoskeleton and inhibiting germ tube formation). Moreover, the increased expression of calmodulin inhibited the antifungal activity of tamoxifen. The strains with mutations in calmodulin are hypersensitive to tamoxifen [54,55]. MYO2 is a calmodulin-binding protein involved in various cellular polarized growth processes [100]. Studies demonstrated that tamoxifen can interfere with the interaction between Myo2p and calmodulin in yeast [55]. Another study also showed that toremifene and tamoxifen exert the anti-cryptococcal activity by directly binding to calmodulin, leading to suppressed calmodulin-mediated calcineurin activation and nuclear localization of the transcription factor Crz1/SP-1 [56,101].

### 4.2. Stilbenoids Targeting Virulence Factors

With the continuous emergence of multidrug-resistant microbes, conventional antibiotics are becoming increasingly ineffective at treating microbial infections. Virulence determines the pathogenicity of pathogens. Pathogens achieve host colonization, tissue damage and immune evasion/modulation with the help of virulence factors. In recent years, there has been a growing interest in disrupting the microbial virulence mechanisms that disarm pathogenic microbes rather than killing them using anti-infective/anti-virulence drugs. Anti-virulence agents aim to inhibit the virulence factors (adhesins, invasins, enzymes and toxins) or other surface proteins that allow the pathogens to survive in adverse conditions, which, in turn, slows the infection rate [102]. It has been greatly encouraged to research and develop agents for clinical use that specifically target the production of virulence factors, without inhibiting planktonic cell growth, to control infection in the resistance era. Firstly, an anti-virulence treatment strategy places less selective pressure on microbes to evolve novel antibiotic resistance mechanisms [103]. Secondly, anti-virulence agents can, in theory, distinguish between the endogenous microbiome and infectious pathogens, which is something that conventional antibiotics do not have.

Surface proteins can assist the bacteria to adhere to the surface of host organ tissues and evade the host’s immune defense with the assistance of transpeptidase [104,105]. For example, the deletion of sortase A (SrtA), an important transpeptidase in *S. aureus*, could downregulate the level of surface proteins and thereby relieve infection symptoms of *S. aureus* in mouse models, without any influence on the bacterial growth [106]. Erianin, a bibenzyl isolated from *Dendrobium chrysotoxum* [107], could inhibit the activity of SrtA at subminimum inhibitory concentrations by binding to SrtA, leading to the reduced adhesion of *S. aureus* to fibrinogen without influencing bacterial growth and thereby improving survival in mice infected with *S. aureus* [57]. 

The toxins have been implicated in the pathogenesis, such as hemolysins, leukocidins, superantigens and surface proteins. For example, α-hemolysin (Hla) is secreted by *S. aureus* and can lyse erythrocyte by directly binding to the cell membrane, forming perforations. Studies have shown that resveratrol and trans-stilbene can markedly inhibit the hemolysis of *S. aureus* at subinhibitory concentrations by repressing the expression of the *hla* gene, thereby attenuating *S. aureus* virulence in the nematode *Caenorhabditis elegans* [58]. The two-component system SaeRS plays a crucial role in the production of virulence factors [108]. Studies showed that resveratrol could reduce the secretion of α-hemolysin by downregulating *saeRS* [59]. Moreover, the toxin RtxA1 of *Vibrio vulnificus* can cause membrane permeabilization upon host contact. It has been shown that although 4,4′-diisothiocyanatostilbene-2,2′-disulfonic acid disodium salt hydrate (DIDS) could not exert antibacterial activity against *V. vulnificus*, it significantly inhibited the secretion of RtxA1 by reducing the expression of *TolCV1* [60]. The outer membrane portal TolC (designated TolCV1 and TolCV2 in *V. vulnificus*), coupled with tripartite efflux pumps, is crucial for the export of RtxA1 [109]. 

Pyocyanin, produced by *Pseudomonas aeruginosa*, can assist bacteria in escaping the host’s immune system and successfully establishing infection by preventing T-cells from effectively responding against *P. aeruginosa* and by inhibiting the activity of monocytes and macrophages, thus reducing the production of cytokines [110]. Therefore, pyocyanin biosynthesis is an important target for novel antimicrobial drug discovery. Pyocyanin synthesis is catalyzed by a series of enzymes encoded by the homologous *phzA1B1C1D1E1F1G1* and *phzA2B2C2D2E2F2G2* operons. The estrogen receptor modulator, raloxifene, was found to inhibit pyocyanin production by binding and inhibiting PhzB2, and thereby attenuating *P. aeruginosa* virulence in a *Caenorhabditis elegans* model [61].

The type III secretion system (T3SS) is a complex “molecular syringe” consisting of a base embedded within the bacterial membrane, a needle filament that connects the bacteria to the host cell, a tip that functions as a platform for the assembly of the translocon, and a transposon that spans the host cell membrane [111]. The T3SS functions to inject effectors of bacteria into host cells, leading to damaged cell structure and compromised immune response and eventually assisting in bacterial infection. The T3SS is an attractive drug target for developing novel antibacterial agents, since it is crucial in virulence and evolutionarily conserved in Gram-negative pathogens. In addition, the endogenous gut microflora lacks the T3SS and therefore is not likely to be affected by anti-T3SS agents. (−)-Hopeaphenol, a natural tetramer of resveratrol, was found to reduce cell entry and subsequent intracellular growth of bacteria by inhibiting the translocation of effector protein YopE from *Y. pseudotuberculosis* and ExoS from *P. aeruginosa* into HeLa cells. (−)-Hopeaphenol could inhibit the expression and secretion of the translocator protein YopD of T3SS [62,63]. The type III secretion system (also called the Hrp secretion system) is encoded by *hrp* (which stands for hypersensitive reaction and pathogenicity) genes in plant pathogenic bacteria. HrpL can activate *hrp* and *avr* gene expression by recognizing consensus motifs in the promotors of these genes [112]. HrpA encodes components of the type III secretion system [113]. Research showed that resveratrol-derived oligomers (hopeaphenol, isohopeaphenol, kobophenol A and ampelopsin A), isolated from grapevine roots, could reduce the pathogenicity of *Pseudomonas syringae* pv. *tomato* DC3000 on tomato leaves by reducing the expression of *hrpA*, *hrpL* and *hopP1* genes without influencing bacterial growth [64,65]. 

### 4.3. Stilbenoids Targeting Biofilms

It is estimated that 40–80% of bacteria and archaea on earth reside in biofilms [114]. A biofilm is a macrocolony of microorganisms that are attached to a surface, and a key factor that cause chronic infections (e.g., cystic fibrosis, dental plaque, chronic wounds, urinary infection, prosthetic joint infection, chronic otitis media, middle ear cholesteatoma, chronic adenoiditis, chronic prostatitis and cardiac valve infection) [115,116]. According to the National Institutes of Health (NIH), more than 80% of human microbial infections are related to biofilms [117]. Biofilms offer microorganisms strong competitive advantages under various environmental challenges [118]. For example, it has been estimated that bacteria in biofilms are 10,000 times more resistant to antibiotics than plankton bacteria [119,120]. Several reasons could account for the antibiotic tolerance of biofilms. Firstly, it is difficult for antibiotics to invade into the deeper layers of biofilm due to the protection of the extracellular polymeric substance (EPS) matrix [121]. In addition, the biofilm increases the transfer rate of horizontal genes, which are responsible for antibiotic resistance [122]. Moreover, the microbes in biofilm have a relatively low rate of cell growth and reproduction. Reduced metabolic activity will lead to decreased antibiotic effectiveness [123]. Moreover, biofilms can also colonize medical devices (for example, catheters and implants). Thus, biofilm is an attractive target for the development of new antibiotics to treat aggressive pathogenic infections.

Biofilm formation can be described in five progressive stages: initial reversible attachment; irreversible attachment; the first layer formation; formation of mushroom- or tower-like structures; and dispersion and reattachment [115]. Diverse studies have shown that stilbenoids can target the major events in the formation processes to prevent biofilm formation and induce mature biofilm clearance.

#### 4.3.1. Prevention of Initial Attachment

It is essential to move close enough to a physical surface for the formation of microbial biofilms. Studies have shown that flagella and fimbriae are crucial for bacterial attachment to surfaces. Stilbenoids can inhibit microbial attachment by reducing these factors. For example, flagella-mediated motility is essential for overcoming the electrostatic repulsion of microbial cells and surfaces, while fimbriae-mediated twitching motility assists bacterial cells to aggregate to form microcolonies [124,125,126]. Resveratrol inhibited flagellin production and the swarming of *Proteus mirabilis*, thereby inhibiting their ability to invade human urothelial cells [66]. Another study showed that resveratrol downregulated the expression of flagella genes, leading to reduced swimming motility and, therefore, compromised adhesion of *Salmonella typhimurium* to HeLa cells [67].

Fimbriae, encoded by the *fimA* gene, can be classified into six genotypes (type I, Ib, II, III, IV and V). Studies have found that resveratrol could suppress the expression of *fimA* and *xadA* (which encode an afimbrial adhesion), thereby reducing cell surface adhesion of the Gram-negative bacterium *Xylella fastidiosa*, which can cause Pierce’s disease in grapevines [127]. Resveratrol dose-dependently prevented the biofilm formation of *P. gingivalis* by reducing the expression of genes which encode fimbriae (type II and IV) [68]. Resveratrol and oxyresveratrol reduced fimbriae production and the swarming motility of uropathogenic *Escherichia coli* (UPEC), leading to reduced biofilm formation and hemagglutinating ability, reducing the defense of UPEC against human whole blood [74]. In addition, type IV toxin-coregulated pilus (TCP) is an important attachment factor [128]. AphB controls the expression of TcpP. TcpP activates TCP by controlling the expression of ToxT, a direct activator of TCP [129]. Resveratrol can suppress biofilm formation of *V. cholerae* by inhibiting the activity of AphB [69]. 

The slime substance polysaccharide intercellular adhesin (PIA) is critical for adhesion on hydrophilic surfaces. It can be synthesized by products of the intercellular adhesion (ica) locus [130]. Studies showed that trans-stilbene could reduce the biofilm by decreasing the expression of the intercellular adhesion locus (*icaA* and *icaD*) in *S. aureus* [58]. 

#### 4.3.2. Targeting Biofilm Maturation

After initial attachment and aggregation, the pathogens will further become embedded and secured in the matrix. This process is called biofilm maturation. Biofilm maturation includes cell–cell interaction, hyphal differentiation (fungi) and extracellular matrix production [131]. The development of the microcolony into a mature biofilm is regulated by various regulatory systems, such as the quorum-sensing (QS) system, the two-component regulatory systems and the type III secretion system [132]. Stilbenoids can target extracellular polymeric substance (EPS), signaling regulatory systems, as well as other surface properties of pathogens that are required for the development of the biofilm. 

Microbes in biofilms are wrapped in a self-produced EPS matrix. The components of the EPS mainly include extracellular DNA, lipids, polysaccharides and proteins [133]. Among them, polysaccharides can provide a number of benefits to the cells in the biofilm, such as assisting in intercellular adhesion and protection and providing structural support [134]. Studies found that resveratrol could inhibit the biofilm formation of *Streptococcus mutans* by reducing the biosynthesis of polysaccharides [70,71]. Water-insoluble glucans form the structural scaffold of biofilms. Extracellular glucosyltransferases GtfB and GtfC are involved in the biosynthesis of water-insoluble glucans. Studies showed that piceatannol could inhibit the activity of GtfB and GtfC, leading to the compromised biofilm formation of *S. mutans* in a dose-dependent manner, both in vitro and in vivo [75]. Another study showed that oxyresveratrol reduced *S. mutans* biofilm formation by suppressing the expression of *gtfB* and *gtfC* [76,135]. 

Acetic acid, as a signaling molecule, can stimulate bacterial biofilm formation [136]. *Streptococcus mutans* is a major inducer of dental caries. An effective way to treat this disease is to replace *S. mutans* by *Lactobacillus casei* within the dental plaque. Studies have found that oxyresveratrol could promote *L. casei* to produce acetic acid, which inhibited the biofilm formation of *S. mutans* and therefore facilitated *L. casei* to compete with *S. mutans* [76,135]. 

Quorum sensing (QS) is a kind of communication between microbial cells. Bacteria can secrete signaling molecules into their surroundings. Signaling molecules can be imported into cells after the bacterial population density reaches a certain threshold, directly binding to the receptor LasR to regulate biofilm formation or the production of virulence factors [83,137]. *Pseudomonas aeruginosa* has four QS systems: *las*, *rhl*, *pqs* and *iqs*. *pqs* QS regulates the biosynthesis of pyocyanin [138]. Pyocyanin can facilitate extracellular electron transfer in *P. aeruginosa* biofilms, therefore supporting the growth of biofilms [139,140]. Research has found that amorfrutin B, a bibenzyl isolated from *Amorpha fruticose* [141], can reduce the pyocyanin production and biofilm formation of *P. aeruginosa* by competitively binding the receptors of the signal molecules ODdHL and PQS, thereby inhibiting the expression of downstream genes such as *lasB*, *rhlA* and *pqsA* [77]. Cajaninstilbene acid analogue 3o also inhibited the QS systems by suppressing the expression of *lasB* and *pqsA*, leading to the inhibition of biofilm formation [78]. Moreover, resveratrol could inhibit the biofilm formation and pyocyanin production of *P. aeruginosa* PAO1 by directly binding LasR [72]. 

Hyphal differentiation is required for robust biofilm formation. Research showed that although mutants of hyphal formation can develop into biofilm, these biofilms are loose, rather than stable, compared with wild-type biofilms [131]. The Ras/cAMP/Efg1 pathway plays a key role in fungal morphological transitions by regulating the expression of hyphae-specific genes, such as *HGC1* (functioning in the polarized growth of hyphae), *ALS3* (encoding a cell wall surface protein related to adhesion), *HWP1* (encoding a cell wall mannose protein, which is essential for the normal growth of the mycelium) and *ECE1* (encoding a membrane protein, which is related to the extension of hyphal). Efg1 is a transcription factor, regulating the expression of hyphae-specific genes [142]. Stilbenoids can target the morphological transition of *C. albicans*, thereby inhibiting biofilm formation. For example, the macrocyclic bisbibenzyl riccardin D has been reported to inhibit the biofilm formation of *C. albicans* both in vitro and in vivo by reducing the expression of hypha-specific genes (*ALS1*, *ALS3*, *ECE1*, *EFG1*, *HWP1* and *CDC35*) and therefore hypha formation [79,80]. Another study showed that pterostilbene could inhibit biofilm formation and destroy mature biofilms both in vitro and in vivo. Pterostilbene could downregulate the expression of filamentation-related genes in the Ras/cyclic AMP (cAMP) pathway, including *ECE1*, *ALS3*, *HWP1*, *HGC1* and *RAS1*. The addition of exogenous cAMP reverted the defect in pterostilbene-induced filamentous growth [81]. 

#### 4.3.3. Disarming Pathogens within the Biofilm

It is clear that biofilms continuously release planktonic cells or small clusters of pathogens by dispersal, seeding new sites of infection and ensuring further replication of biofilms [143]. Stilbenoids can interfere with this process by targeting the enzymes involved in it, thereby eradicating the preformed biofilms. For example, *S. aureus* could escape neutrophil extracellular traps by cleaving their DNA-backbone through secreting micrococcal nuclease (MN). MN also inhibits biofilm development and adhesion by cleaving the extracellular DNA. Reports have shown that resveratrol could reduce MN production, which assists the host immune system to clear the biofilm of *S. aureus* in clinical situations [73].

### 4.4. Reversing Antibiotic Resistance

Antimicrobial resistance is a global health problem. Microbes use various mechanisms to resist the action of antimicrobials [144]. Some are intrinsic mechanisms, by which bacteria can use genes they already possess to avoid antibiotic exposure, and some are acquired mechanisms, by which bacteria evolve new genetic materials under antibiotic exposure in order to survive. Inhibiting resistance mechanisms is a potential approach to combat drug resistance.

Studies have shown that ATP synthase is an intrinsic resistance factor of bacteria. For example, the inactivation of genes encoding the subunits of ATP synthase could significantly increase the efficacy of gentamicin against *S. aureus* [145]. Another study showed that the deletion of the ATP synthase subunit *atpG* increased the efficacy of aminoglycosides against *E. coli* [146]. Resveratrol could enhance the efficacy of aminoglycosides against various Gram-positive pathogens. Mechanism research has revealed that resveratrol could inhibit the activity of ATP synthase [82]. 

Biofilm reduces antibiotic permeability due to the protection of the EPS matrix. For example, the minimum biofilm eradication concentrations of clinical *Acinetobacter baumannii* isolates were 44, 407 and 364 times higher than the minimum bactericidal concentrations for colistin, ciprofloxacin and imipenem, respectively [147]. Considering the roles of biofilm in antimicrobial resistance, inhibiting the formation of biofilm theoretically enhances the effect of antibiotics. *P. aeruginosa* infection is a serious disease in patients with cystic fibrosis and is difficult to treat due to the persistence of biofilms. A study has shown that the biofilms of *P. aeruginosa* PAO1 were more susceptible to aminoglycosides in the presence of resveratrol. Mechanism research has revealed that resveratrol could function as a QS inhibitor. QS deficiency leads to thin and less developed biofilms [83]. Resveratrol could inhibit the expression of the signaling molecule synthase genes *lasI* and *rhlI* in *P. aeruginosa* PAO1 biofilms [84]. 

Antimicrobial resistance can be conferred by drug-modifying enzymes. The enzymatic degradation of antibiotics involves the hydrolysis of their functional groups, thereby rendering them ineffective. For example, the New Delhi metallo-β-lactamase-1 (NDM-1) can hydrolyze almost all β-lactam antibiotics. Liu et al. discovered that pterostilbene could significantly inhibit the NDM-1 hydrolysis activity by binding to the catalytic pocket of NDM-1, thereby preventing the substrate from binding to this protein, effectively restoring the effectiveness of meropenem against NDM-expressing strains and protecting mice from pneumonia caused by *Klebsiella pneumoniae* [85]. 

Bacteria can prevent antibiotics from reaching to corresponding target by enzymatic modification of the target protein. For example, polymyxin disrupts membrane integrity by binding to the lipid A moiety of lipopolysaccharides (LPS), ultimately causing cell death. Polymyxin is the last line of defense against Gram-negative bacterial infections. However, the plasmid-borne mobilized colistin resistance gene *mcr-1* (encoding phosphoethanolamine transferase) significantly decreases the efficiency of polymyxin against Gram-negative bacteria. *mcr-1* is widely spread throughout the world and significantly threatens the usefulness of polymyxin. It can append phosphoethanolamine to a phosphate on lipid A, reducing the electrostatic interaction between LPS and polymyxin, thereby increasing bacteria resistance to polymyxin [148]. Recent studies have demonstrated that Cajaninstilbene acid from *Cajanus cajan* (L.) Millsp can significantly restore the susceptibility of polymyxin B to *mcr-1* positive Gram-negative bacteria through inhibiting the enzymatic activity of MCR-1 via binding the active center [86]. 

Active efflux is mediated by the upregulation of efflux pumps that extrude antibiotics from bacteria to reduce drug concentrations in the cell to non-cytotoxic levels. Recently, efflux pump inhibitors (EPIs) that can reverse antibiotic resistance have received great attention. The AcrAB-TolC multidrug efflux pump, consisting of the inner membrane protein AcrB and the outer membrane protein TolC, linked by AcrA, can transport various toxic compounds out of bacteria, contributing to drug resistance [149,150]. Research has shown that resveratrol could significantly inhibit *tolC*-promoter activity and decrease *tolC* expression [87]. Moreover, stilbenes (resveratrol, pterostilbene and pinosylvin) enhance the susceptibility of *A. butzleri* strains to chloramphenicol, erythromycin and ciprofloxacin by acting as EPIs [88,89]. Proton motive force (PMF) is essential for efflux pumps to export drugs [151]. Piceatannol could decrease PMF of *S. aureus*, particularly the Δψ component, leading to increased sensitivity of bacteria to ciprofloxacin [90]. 

## 5. Conclusions

The heavy use of antibiotics to treat pathogenic infections in humans and animals, as well as in agriculture has promoted the development of drug resistance in microbial populations. Presently, we are entering an era of ever-increasing resistance, in which the treatment of pathogen infection through the use of some conventional antibiotics can no longer be taken for granted, and the need for new antibiotics has never been more urgent [152]. The development of effective therapies against resistant microorganisms and emerging pathogenic microorganisms is a public health priority. The chemical structure of stilbenoids is new in the field of antimicrobial drugs and has received great attention. Stilbenoids demonstrate a diverse range of functions, including direct microbicidal activity, a wide range of effects on biofilms, virulence factors and the reversal of drug resistance, highlighting the considerable possibility of developing stilbenoids as a potential antimicrobial agent. Drug repurposing has emerged as a novel paradigm to find antimicrobial agents. Several stilbenoids with antimicrobial activities have been approved for clinical use, such as toremifene, raloxifene and resveratrol, which are easier to obtain approval for in the treatment of pathogenic infections. However, the practical application of stilbenoids needs to consider challenges such as toxicity, solubility, stability or other factors in advance. Some of these problems could be addressed by chemical modification or by developing delivery devices such as nanocapsules, liposomes and nano-carrier systems [153]. We expect major breakthroughs and revolutionary technologies in the research on stilbenoids. This will serve as a new driving force for improving people’s health. 

## Figures and Tables

**Figure 1 pharmaceutics-16-00663-f001:**
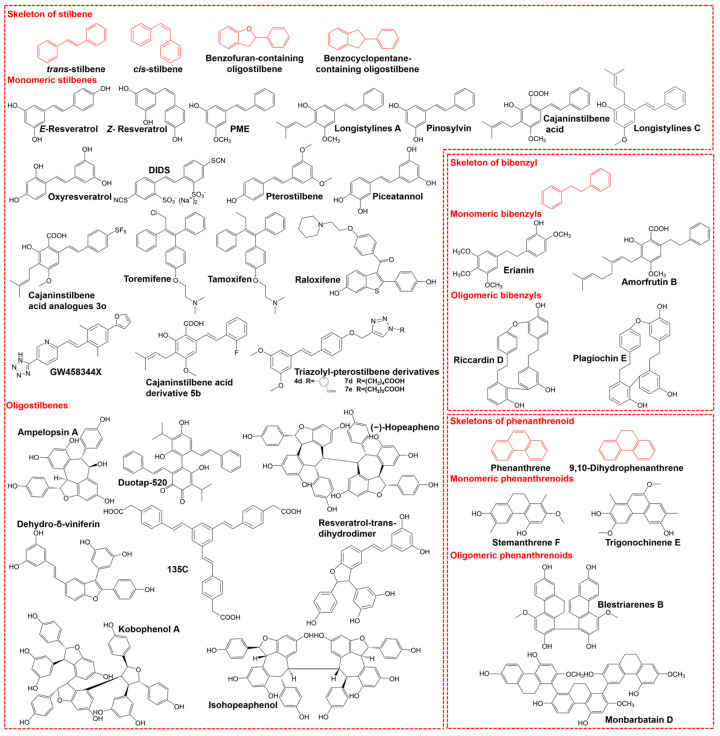
Skeleton characteristics and representatives of three kinds of stilbenoids. Stilbenoids can be divided into three categories according to their chemical structure, including stilbene, bibenzyl and phenanthrenoid.

**Figure 2 pharmaceutics-16-00663-f002:**
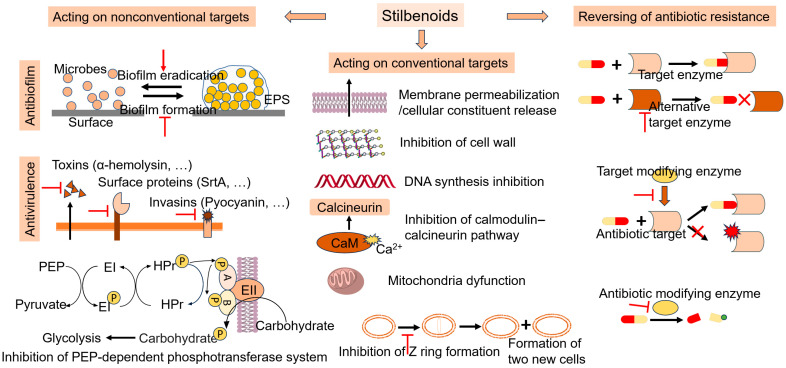
Overview of the antimicrobial mechanisms of stilbenoids. The red arrow represents activation. The red T represents inhibition. The red cross means the reaction is blocked.

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
