# Peer review of "Progress of Antimicrobial Mechanisms of Stilbenoids"

_pharmaceutics, 2024, doi:10.3390/pharmaceutics16050663_

Round 1

Reviewer 1 Report

Comments and Suggestions for Authors

The article is a comprehensive review of the  of direct antimicrobial properties, antibiofilm and antivirulence activities, to the reversing properties of drug resistance that stilbenoids show. The continuous chapters are well planned and thoroughly discussed. The problem of resistance of pathogenic microorganisms is very topical. Stilbenoids are natural compounds with proven antimicrobial properties, therefore the an in-depth understanding of their antimicrobial activity could lead to the design of new drugs.

I recommend the manuscript for publication after some corrections:

-          (1) Sars and Covid – they are illness, not pathogens; please correct

-          (2) Figure 2. – too long title; the title should be shortened and the discussion should be in the main text; If the figure is based on some other figures taken from the literature, it should be written [based on …].

-          (3) A table with gathered antimicrobial parameters (e.g. MICs or others) for selected stilbenoids should be added; it will show at the beginning the antimicrobial potential of these compounds,

-          (4) different way of writing the names of molecules, e.g. e Lipid II or lipid II; TagG, TagH or tagH, tagA and tag; Cajanin Stilbene Acid – small letters; SaeRS or saeRS; please correct it.

-          (5) lack of the formula of the discussed compounds, e.g. Duotap-520, aza-stilbene GW458344X, tris-stilbene 135C, Plagiochin E; 4,4'-diisothiocyanatostilbene-2,2'-disulfonic acid disodium salt hydrate; raloxifene; Hopeaphenol; (hopeaphenol, isohopeaphenol, kobophenol A and ampelopsin A, amorfrutin B, Cajaninstilbene acid analogue 3o, riccardin D, pterostilbene etc. – most of them should be included in the manuscript

-          (6) in vitro and in vivo – text once written in italic or straight font.

Comments on the Quality of English Language

The work is written correctly. Still, I recommend a thorough read and slight linguistic corrections.

Author Response

We are very grateful to the reviewer for the recognition of this article, and at the same time thank the reviewer for the valuable comments. We have made the following modifications according to the comments:

 (1) Sars and Covid – they are illness, not pathogens; please correct;

Reply: we have revised “SARS” and “COVID-19” to “SARS-CoV” and “SARS-CoV-2”, respectively.

 (2) Figure 2. – too long title; the title should be shortened and the discussion should be in the main text; If the figure is based on some other figures taken from the literature, it should be written [based on …].

Reply: we have moved the discussion to the main text.

 (3) A table with gathered antimicrobial parameters (e.g. MICs or others) for selected stilbenoids should be added; it will show at the beginning the antimicrobial potential of these compounds,

Reply: we have added a table to gather the antimicrobial parameters.

 (4) different way of writing the names of molecules, e.g. e Lipid II or lipid II; TagG, TagH or tagH, tagA and tag; Cajanin Stilbene Acid – small letters; SaeRS or saeRS; please correct it.

Reply: we have corrected these mistakes.

 (5) lack of the formula of the discussed compounds, e.g. Duotap-520, aza-stilbene GW458344X, tris-stilbene 135C, Plagiochin E; 4,4'-diisothiocyanatostilbene-2,2'-disulfonic acid disodium salt hydrate; raloxifene; Hopeaphenol; (hopeaphenol, isohopeaphenol, kobophenol A and ampelopsin A, amorfrutin B, Cajaninstilbene acid analogue 3o, riccardin D, pterostilbene etc. – most of them should be included in the manuscript.

Reply: we have added the formula of the mentioned compounds in Fig. 1.

 (6) in vitro and in vivo – text once written in italic or straight font.

Reply: we have corrected these mistakes.

Reviewer 2 Report

Comments and Suggestions for Authors

The present manuscript is a comprehensive review about the mode of action of stilbenoids from view of direct antimicrobial properties, antibiofilm and antivirulence activities, to the reversing of drug resistance. The review is well written and exhaustive, thus in my opinion it can be published on Pharmaceutics after minor revision.

- Same typos are present, for example some spaces are missing. Please double check the manuscript.

- The authors reviewed the different ways the stilbenoids can interact with living cells. I suggest them to add a table to summarize all the discussed modes to give to the reader at a glance the focus of their manuscript.

-the authors could better focus the attention on the structure-activity relationship of the discussed stilbenoids

Comments on the Quality of English Language

English is fine

Author Response

We are very grateful to the reviewer for the recognition of our work, and at the same time, we would like to thank the reviewer for the constructive comments:  

(1) Same typos are present, for example some spaces are missing. Please double check the manuscript.

Reply: We have reviewed the errors in this article in detail and have made changes.

(2) The authors reviewed the different ways the stilbenoids can interact with living cells. I suggest them to add a table to summarize all the discussed modes to give to the reader at a glance the focus of their manuscript.

Reply: We have added a table to summarize parameters of the discussed compounds.

(3) the authors could better focus the attention on the structure-activity relationship of the discussed stilbenoids.

Reply: The structure-activity relationship has been added. 

Reviewer 3 Report

Comments and Suggestions for Authors

The present manuscript by Xiancai Li et all is an extensive and well written review of antimicrobial mechanisms of stilbenoids. Overall it could be of interest to researchers in the field and I recommend its publication without revisions.

Author Response

Many thanks to the reviewer for the high evaluation of our work.

Reviewer 4 Report

Comments and Suggestions for Authors

The work has some value, but needs work. Reduce the rațio of similarity on the report.

The manuscript presented for evaluation presents a literature study on stilbenoids and their antimicrobial effects. The subject of the review is not very original, but the manuscript seems to add somehow a new perspective. Still, the authors should cite similar review work and highlight to the readers what does their work adus new and its strenghts. See for example review that could be mentioned:

  1. Resveratrol and Other Natural Oligomeric Stilbenoid Compounds and Their Therapeutic Applications
  2. Stilbenoids as Promising Natural Product-Based Solutions in a Race against Mycotoxigenic Fungi: A Comprehensive
  3. Stilbenoids: A Natural Arsenal against Bacterial Pathogens

The work presented here is not consistent.
The authors mention that stilbenoids are plant products, but they discuss many synthetic compounds or tamoxifen. These compounds should not be the subject of the paper. On the other hand, there are many stilbenoids that are not discussed. There should be a structured activity discussion, something like the influence of the number of phenolic groups, or something like that.

Another advise it would be to have a critical approach of the data. The review should not be just a collection of data, it should be a critical analysis pointing out the things that are not yet known and need more research and the things that are controversial. Are there studies that seems to be incorect? Or misleading?

Comments on the Quality of English Language

Ok, but needs editing.

Author Response

We agree with those suggestion. We will be happy to edit the text further based on the helpful comments from review. Many thanks to the reviewer for the valuable comments:

(1) Reduce the rațio of similarity on the report.

Reply: We have modified the language of the article to reduce the ratio of similarity.

(2) The manuscript presented for evaluation presents a literature study on stilbenoids and their antimicrobial effects. The subject of the review is not very original, but the manuscript seems to add somehow a new perspective. Still, the authors should cite similar review work and highlight to the readers what does their work adus new and its strenghts. See for example review that could be mentioned:

Reply: We have cited the references mentioned by the reviewer and stated in detail the differences between our article and the work of others in introduction part.

(3) The work presented here is not consistent.

The authors mention that stilbenoids are plant products, but they discuss many synthetic compounds or tamoxifen. These compounds should not be the subject of the paper.

Reply: We have revised the ambiguous language of the article, and this article does not restrict the source of the compound.

(4)On the other hand, there are many stilbenoids that are not discussed. There should be a structured activity discussion, something like the influence of the number of phenolic groups, or something like that.

Reply: Some stilbenoids have a repetitive antimicrobial mechanism and therefore are not mentioned here. The structure-activity relationship has been added. 

(5) Another advise it would be to have a critical approach of the data. The review should not be just a collection of data, it should be a critical analysis pointing out the things that are not yet known and need more research and the things that are controversial. Are there studies that seems to be incorect? Or misleading?

 Reply: The main purpose of our work is to describe the antimicrobial mechanisms.